# Understanding Renal Tubular Function: Key Mechanisms, Clinical Relevance, and Comprehensive Urine Assessment

**DOI:** 10.3390/pathophysiology32030033

**Published:** 2025-07-03

**Authors:** Mario Alamilla-Sanchez, Miguel Angel Alcalá Salgado, Victor Manuel Ulloa Galván, Valeria Yanez Salguero, Martín Benjamin Yamá Estrella, Enrique Fleuvier Morales López, Nicte Alaide Ramos García, Martín Omar Carbajal Zárate, Jorge David Salazar Hurtado, Daniel Alberto Delgado Pineda, Leticia López González, Julio Manuel Flores Garnica

**Affiliations:** 1Department of Nephrology, Centro Médico Nacional “20 de Noviembre”, Colonia del Valle, Mexico City 03104, Mexico; 2Department of Nephrology, Hospital Christus Muguerza, Saltillo 25204, Mexico

**Keywords:** tubular function, renal physiology, tubulopathy, urine biomarker

## Abstract

Renal function refers to the combined actions of the glomerulus and tubular system to achieve homeostasis in bodily fluids. While the glomerulus is essential in the first step of urine formation through a coordinated filtration mechanism, the tubular system carries out active mechanisms of secretion and reabsorption of solutes and proteins using specific transporters in the epithelial cells. The assessment of renal function usually focuses on glomerular function, so the tubular function is often underestimated as a fundamental part of daily clinical practice. Therefore, it is essential to properly understand the tubular physiological mechanisms and their clinical association with prevalent human pathologies. This review discusses the primary solutes handled by the kidneys, including glucose, amino acids, sodium, potassium, calcium, phosphate, citrate, magnesium and uric acid. Additionally, it emphasizes the significance of physicochemical characteristics of urine, such as pH and osmolarity. The use of a concise methodology for the comprehensive assessment of urine should be strengthened in the basic training of nephrologists when dealing with problems such as water and electrolyte balance disorders, acid-base disorders, and harmful effects of commonly used drugs such as chemotherapy, antibiotics, or diuretics to avoid isolated replacement of the solute without carrying out comprehensive approaches, which can lead to potentially severe complications.

## 1. Introduction

It is widely accepted that most species originated from marine ancestors. The ability of fish to survive through all stages of evolution has a significant impact on ecosystems [1]. Hydrosaline regulation is essential for adapting reptiles and mammals to terrestrial environments [2].

Interestingly, certain fish species, despite lacking glomeruli, have a cellular system that can effectively excrete substances such as creatine, creatinine, and phenolsulfonphthalein [3]. In the hyperosmotic environment of the sea, solutes lost by renal excretion are efficiently recovered by the branchial system, particularly NaCl. In freshwater, fish employ a different strategy: removing water by osmosis and gaining NaCl through a diffusion mechanism [1]. This intricate interplay of mechanisms underscores the role of tubular function as part of the adaptive complex that has enabled evolution ‘from fish to philosopher.’

Tubular adaptive processes can be classified as functional or structural under conditions of chronic impairment of renal function. An enlargement of the tubular cells, particularly the proximal tubule, is observed along with a significant increase in the single nephron glomerular filtration rate (SNGFR) [4,5] and flow resorption rates of up to 60%, secondary to the effect of glomerulotubular balance related to increased transepithelial potential difference of the remaining functional nephrons [6]. Functional nephrons are impermeable to water in the distal tubules in the absence of vasopressin. However, under conditions of high vasopressin concentrations, the tubular capacity of urine may be severely depressed [7], which produces characteristic isosthenuria in patients with chronic kidney disease.

It is assumed that tubular function deteriorates alongside glomerular function, a process called the ’intact nephron hypothesis’ [8]. However, the clinical experience of most nephrologists has observed that patients with advanced renal impairment, even those with chronic kidney support therapy, have a completely different course according to their ability to produce urine, and classic monitoring parameters such as phosphate, potassium, and interdialytic gain can be diametrically opposed. This phenomenon occurs due to the dissociation between the decline in glomerular filtration and tubular secretion. Transporters like MRP2/4, BCRP, OAT1-3, OCT2, MATE 1/2K, and URAT1 facilitate the excretion of organic solutes that significantly improve uremic toxemia and can improve long-term cardiovascular prognosis [9,10], even when GFR is severely depressed.

In recent years, exceptional progress has been made in functional monitoring. Biomarkers play a crucial role in early detection and monitoring of organ diseases. This review explores urine solutes, which serve as functional markers that have been valuable in clinical assessments, along with their diagnostic and prognostic relevance in applicable clinical situations. However, structural biomarkers are beyond the scope of this review. We employed a structured search strategy using a combination of Medical Subject Headings (MeSH) and text keywords related to tubular function and solute handling in the kidney, focusing on articles published since the 2000s. The search terms included: “kidney tubular function” OR “renal tubule physiology”, AND “urinary biomarkers” OR “solute handling” OR “tubulopathy”, AND “acute kidney injury” OR “chronic kidney disease” OR “AKI” OR “CKD”, AND individual solutes and physiological processes such as “glucosuria”, “phosphaturia”, “aminoaciduria”, “uricosuria”, “citraturia”, “magnesuria”, “calciuria”, “urine osmolality”. Classical or foundational studies were included when no recent high-quality evidence was available regarding specific transport mechanisms.

## 2. Proximal Tubule Involvement

The proximal tubule is essential for maintaining the balance between salt and water, where most of the filtered solutes are reabsorbed. It displays high metabolic activity and significant tubular secretion, which affect the elimination of drugs and other solutes that are not easily filtered through the glomerular filtration barrier.

Proximal tubule cells establish connections through low-resistance junctional complexes, facilitating solute passage via the paracellular shunt pathway. Acidification occurs through an Na/H antiporter located in the luminal membrane and a chloride-independent, voltage-dependent bicarbonate exit mechanism in the basolateral membrane. The reabsorption of NaCl is comprised of approximately 40% passive reabsorption and 60% active reabsorption. The lumen-negative potential difference influences chloride absorption. Conversely, in convoluted tubules, active transport is neutral, with equivalent sodium and chloride transport occurring across cells [11].

A more detailed description of the specific solute reabsorption in this segment of the nephron is provided in each section.

### 2.1. Glucosuria

A classical study conducted by Butterfield et al. showed that the renal tubular reabsorption threshold can range from 130 mg/dL to >200 mg/dL and may increase with age [12]. However, an average Tmax value of approximately 180 mg/dL is commonly accepted [13]. The sodium-glucose co-transporters type 1 and 2 (SGLT) reabsorbs 90% and 10% of glucose in the proximal tubule. SGLT2 (SLC5A2) is expressed in the S1 and S2 segments, while SGLT1 is expressed in the S3 segment. SGLT2 transports an equimolar 1:1 quantity of sodium and glucose, and SGLT1 (SLC5A1) transports two molecules of sodium for each glucose [14,15]. Familial renal glucosuria is related to mutations in the SCLC5A2 gene [16].

Glucosuria is a functional test that provides evidence for tubular injury. Some studies have been conducted in patients with acute kidney injury (AKI) and glucosuria without diabetes. Tietäväinen et al. analyzed urine glucose in 195 patients with hantavirus infection, found a prevalence of normoglycemic glycosuria in 12% of patients at the time of hospital admission. They detected a positive association between glycosuria and AKI severity, with higher blood urea concentrations, albuminuria, and microscopic hematuria. It should be noted that at follow-up, 96% of patients reversed their dipstick glycosuria [17].

Many nephrotoxins are known to be related to direct tubular injury. For example, light chains can be toxic when they accumulate in the cytoplasm and interact with intracellular organelles. In fact, the presence of complete Fanconi syndrome in adults over 40 years of age should raise suspicion of the coexistence of monoclonal gammopathy. Proximal light chain tubulopathy (PLCT) is characterized by cytoplasmic light chain inclusions within proximal epithelial cells, with kappa chains being the most frequently involved (85%). Despite normoglycemia, it is characterized by low-grade proteinuria, slowly worsening renal function, and loss of electrolytes and glucose in urine [18,19,20]. Therefore, it is relevant to study at least one urinary dipstick to rule out tubular dysfunction in all patients with monoclonal gammopathy.

The interaction between glomerular and tubular functions is evident. Moreover, most lesions with documented tubular dysfunction have variable degrees of glomerular lesions and vice versa. Various studies have investigated the association between glucosuria and histological or clinical findings. A glomerular lesion pattern of nodular glomerulosclerosis without diabetes with proteinuria in the nephrotic range and glycosuria has been reported [21]. Likewise, acute tubulointerstitial nephritis (ATIN) induced by antibiotics has been reported in the presence of AKI and isolated intense glycosuria that resolved after withdrawal of antibiotics and prescription of high doses of prednisone. This effect can be explained by direct and isolated damage to SGLT2 [22]. According to Lee et al., in a study conducted on 28 patients with ATIN, 68% of patients had glycosuria, which was more frequent than hypophosphatemia (18%), hypouricemia (18%), and hypokalemia (18%), compared with only 6% of 116 patients with other etiologies of kidney injury; glycosuria had a sensitivity of 68%, specificity of 94%, LR+ of 11.2, and LR- of 0.34 for the diagnosis of ATIN [23].

Ormond et al. conducted a 24-month retrospective study of 115 patients without diabetes. The prevalence of glucosuria was 10%, with membranous nephropathy the most associated; also, patients with normoglycemic glucosuria had higher serum creatinine, higher albuminuria, and lower serum albumin levels [24]. More recently, Liu et al. retrospectively analyzed 1313 patients with membranous nephropathy, detecting a glucosuria prevalence of 10.8%. Notably, the non-remission ratio of proteinuria was 45% in the glucosuria group compared to 12.5% in the control group, and the complete response ratio was 19.7% vs. 63.5% in the glucosuria group and the control groups, respectively. Also, it was more probable that the glomerular filtration rate dropped 50% from baseline in the glucosuria group at follow-up, suggesting that the proximal tubular injury marker was closely related to clinical prognosis in these subgroups of patients [25].

Paradoxically, Fishman et al., in a retrospective study that included more than 2 million conscripts from the Defense Force, reported that normoglycemic glucosuria had a prevalence of 0.04%, more frequent in males than females. The presence of glucosuria had an adjusted odds ratio for overweight and obesity of 0.66 (95% CI 0.50–0.87) and 0.62 (95% CI 0.43–0.88), respectively; furthermore, adolescents with normoglycemic glucosuria had an odds ratio for systolic blood pressure 130–139 mmHg of 0.74 (95% CI 0.60–0.90), suggesting a protective role of normoglycemic glucosuria in a subset of population [26]. It is possible that the effect can be compared with reports that patients treated with SGLT2 inhibitors may lose up to 300 kcal/day and can be associated with a slight reduction in systolic and diastolic blood pressure [27,28,29,30]. However, these data are inconclusive and require external validation to establish plausible causal associations. See Figure 1.

### 2.2. Aminoaciduria

Amino acids freely pass through the glomerular filtration barrier, but only trace amounts are detected in urine. This process involves secondary active transport in proximal tubular cells for reabsorption. When a genetic disorder is present, the disease can be diagnosed during childhood. Normal children excrete approximately 2.5 mg/kg/day of total amino acids, and in the context of genetic disorders, the loss of amino acids may exceed normal values by 50-fold [31]. The Dent-Walsh classification of aminoaciduria into three types is intuitive and attempts to explain the primary phenomena: overflow aminoaciduria, renal aminoaciduria, and unclassified aminoaciduria [32]. However, most cases of the unclassified type may be related to an acquired renal aminoaciduria phenotype. In addition, some cases of overflow aminoaciduria need to be explained, along with defects in renal capture or metabolism of amino acids in proximal tubular cells.

More than 80% of the amino acids filtered by the glomerulus are neutral [33]. Approximately 50 g of amino acids are filtered daily in healthy adults [34]. B0AT1 (SLC6A19) and B0AT3 (SLC6A18) transporters have variable affinities for all neutral amino acids and consist of a sodium-coupled cotransporter expressed at the S1/S2 brush border of the proximal tubule and in the small intestine [35]. The B0AT1 defect causes Hartnup syndrome, an autosomal recessive disorder that results in the loss of neutral amino acids [36]. SLC6A20, also known as SIT1 or IMINOB, is a protein that co-transports L-proline, 2 sodium ions, and 1 chloride ion [37,38]. B0AT1, B0AT3, and IMINO require transmembrane protein collectrin (Tmem27) to stabilize and promote the expression of the transporter at the brush border [39]. The co-transport of glycine and hydrogen corresponds to PAT1 and PAT2 (SLC36A1 and SLC36A2) [40]. EAAC1 (SLC1A1) reabsorbs acidic amino acids, whereas rBAT (SLC7A9) reabsorbs basic amino acids and cysteine in exchange for a neutral amino acid [41,42]. On the basolateral side, LAT2 (SLC7A8) and SNAT3 (SLC38A3) transporters export or import glutamine, respectively, while LAT1 (SLC7A7) exports glutamine in exchange for neutral amino acids [40,43].

In early experiments, aminoaciduria was studied as an early marker of tubular injury in models of gentamicin- and d-serine-induced nephrotoxicity [44,45]. Pathologies related to acquired or hereditary Fanconi syndrome should be ruled out as a specific indication of proximal tubular dysfunction, especially glucosuria, renal phosphate wasting, and lowweight proteinuria. Inherited conditions include cystinosis, hereditary fructose intolerance, tyrosinemia type I, Lowe’s syndrome, Dent’s disease, lysinuric protein intolerance, galactosemia, Wilson’s disease, and mitochondrial myopathies [46,47]. Secondary Fanconi syndrome is related to various substances, including chemotherapy drugs such as platinum and Ifosfamide, antiviral medications like tenofovir and cidofovir, anticonvulsants such as valproic acid and topiramate, antibiotics such as gentamicin and tetracyclines, heavy metals like lead and cadmium, dysproteinemias, aristolochic acid, toluene, and autoimmune diseases such as Sjögren syndrome [46,48].

Extensive research has analyzed the relationship between metabolomics and clinical outcomes, with a focus on the progression of diabetes. Many of the metabolites analyzed were amino acids found in urine. Bingham et al. evaluated the urinary excretion of 16 amino acids in patients with HNF-1a mutations and age-matched non-diabetic control subjects, type 1 (T1D) and type 2 diabetes (T2D) patients, and patients with diabetes and CKD. The researchers found that patients with HNF-1a mutations had a reduced renal glucose threshold and generalized aminoaciduria that was not related to albuminuria, but was accompanied by glucosuria. These results suggest that glucosuria might cause depolarization and dissipation of the electrical gradient of sodium-dependent amino acid transporters located in proximal tubular cells, leading to aminoaciduria [49].

A study conducted by Melena et al. evaluated 40 patients with diabetic ketoacidosis and T1D. This study showed that the urinary concentrations of histidine, threonine, tryptophan, and leucine were consistently higher during the first 8 h of sampling and substantially reduced at 3 months. This suggests that proximal tubular dysfunction may play a pathogenic role, which is still largely unexplored in patients with T1D who have DKA [50].

In other study conducted by Wang et al. analyzed the metabolic profiles of amino acids in patients with diabetic kidney disease (DKD). The study found an inverse association between the serum levels of histidine and valine and their urinary levels in patients with DKD compared to patients with T2D but without DKD or healthy controls. The researchers generated a diagnostic model with an AUC of 0.85 and an accuracy of up to 92% based on amino acid profiles in plasma, urine, and saliva, which are considerably different in patients with DKD. These results suggest an altered amino acid metabolic profile is associated with the progression of DKD [51].

In an interesting study, Weng et al. found that a combination of three amino acids composed of branched-chain amino acids (valine, leucine, isoleucine) or aromatic amino acids (tyrosine and phenylalanine) can be used as biomarkers to predict the risk of diabetes, with a more than fivefold higher risk for individuals in the top quartile [52].

Pena et al. followed 90 patients with T2D for 2.9 years from the PREVEND study and Steno Diabetes Center and divided them into two groups: patients with normo-microalbuminuria and patients with micro-macroalbuminuria, both matched with controls, and metabolomic analysis was performed on plasma and urine. Both urine glutamine and tyrosine were found to be associated with the progression of microalbuminuria to macroalbuminuria, being specific for diabetes because urine amino acids were not different between patients with hypertension and controls without type 2 diabetes [53].

In a recent study by Luo et al., a combination of urinary amino acids (tyramine and phenylalanylproline) showed excellent diagnostic performance, with an AUC of 0.94 in patients with T2D and DKD. Additionally, a combination of 60 urinary metabolites has been linked to the downregulation of OAT1, which plays a crucial role in the solute regulation of proximal tubule physiology [54]. These data suggest that abnormal metabolic profiles, including urinary amino acids, play an important role in predicting or progressing DKD.

Finally, Vergara et al. found that patients with AKI and COVID-19 who had elevated levels of urinary angiotensin-converting enzyme 2 (ACE2) were associated with increased aminoaciduria [55]. ACE2 is homology to collectrin/Tmem27 in its membrane-anchoring domain; therefore, it interacts indirectly with B0AT1 [56].

### 2.3. Phosphaturia

Phosphate is present in most foods consumed worldwide, and with the increase in the consumption of foods with additives, the phosphate content of the diet has increased proportionally [57]. In fact, additives increase phosphate content by as much as 70% [58]. The gut, kidneys, and bones control phosphate homeostasis in interconnected multi-organ networks. Although the gut is essential for absorption, its regulation is poor. Therefore, many of the most important events for maintaining normal serum phosphate concentrations occur at the renal level.

Products with additives can be absorbed more than 90% of the intestinal tract [57], with the apical transporters NaPi2b (SLC34A2) and Pit-1/2 (SLC20A1/2) involved in transcellular transport. However, paracellular transport is even more important for enterocyte absorption [59].

In the renal tubules, phosphate reabsorption is mainly regulated by three transporters: NaPi2a (SLC34A1), NaPi2c (SLC34A3), and Pit-2 (SLC20A2) [60]. Type II transporters (NaPi) have a preference for divalent phosphate (HPO_4_^2-^) and are electrogenic with a 3:1/2:1 stoichiometry for NaPi2a and NaPi2c, respectively; Type III transporters (Pit) transport monovalent phosphate (H_2_PO_4_^−^) more efficiently with a stoichiometry 2:1 Na:H_2_PO_4_^−^ [61].

NaPi2a/c is responsible for the reabsorption of 80% of filtered phosphate and is located predominantly in the S1 segment of the proximal tubule [62]. The main regulators of protein expression include PTH, FGF-23, dietary phosphorus, dietary magnesium, and metabolic acidosis [60]. Pit-2 is responsible for nearly 20% of filtered phosphate, seems regulated by dietary phosphate, and is restricted to the S1 segment on a normal diet, but can be expressed in all proximal tubular segments when dietary phosphate is low [63]. The basolateral transporter for phosphate exit remains elusive; however, it may occur through xenotropic and polytropic retroviral receptors (XPR1) [64,65].

As proximal tubular cells reabsorb more than 80% of filtered phosphate, phosphaturia is a characteristic clinical sign of proximal tubular dysfunction. Biochemical evaluation includes 24-h urinary phosphate (24-hP), fractional excretion of phosphate (FEP), tubular reabsorption of phosphate (TRP), and the ratio of tubular maximum reabsorption of phosphate to GFR (TmP/GFR).

Recent studies have analyzed functional changes in the process of tubular phosphate resorption in the context of systemic diseases.

Emmens et al. analyzed the TmP/GFR in 2085 patients with heart failure and found that 67% had a low TmP/GFR (<0.8 mmol/L). They observed that patients with lower TmP/GFR had more advanced heart failure, lower eGFR, elevated levels of NGAL, and were associated with a higher risk of all-cause mortality (HR 2.8; 95% CI 1.37–5.73, *p* = 0.005), independent of eGFR [66].

From the nephrotoxicity context, Waheed et al. analyzed 15 HIV-infected patients in treatment with tenofovir and elevated FEP (mean: 34%) in whom renal function deteriorated from eGFR 104 ± 17 mL/min/1.73 m^2^ to 69 ± 19 mL/min/1.73 m^2^ at the time of tenofovir discontinuation (mean: 64 months). In patients with repeated FEP after tenofovir discontinuation, 9 experienced improvement in FEP [67]. These data highlight the importance of continued monitoring in patients taking drugs with known tubular toxicity, even with apparently normal eGFR.

In another interesting study, researchers evaluated pre-donation TmP/GFR in 165 kidney donors 12 months after transplantation and found that pre-donation TmP/GFR was associated with recipient eGFR after 12 months (GFR 6 mL/min lower per 1 mg/dL decrement of TmP/GFR), but no association was detected between tubular biomarkers (NGAL, KIM-1) and recipient GFR [68], highlighting the importance of functional markers over injury biomarkers in predicting renal function in this subset of patients.

Finally, 191 patients were included in a study performed to determine if phosphaturia itself is related to nephrotoxicity and disease progression in patients with CKD (mean eGFR 19 mL/min/1.73 m^2^); the researchers evaluated phosphate load according to 24-hP excretion per creatinine clearance (24-hP/CrCl) and stratified by quartiles adjusting for multiple variables. Patients with 24-hP/CrCl in quartiles 2, 3, and 4 vs. 1 had more risk for the composite outcome (ESKD or sustained 50% reduction in eGFR), with adjusted HR: 3.07 (0.97–11.85), 7.52 (2.13–32.69), and 7.89 (1.74–44.33), respectively [69]. These data emphasize the known role of tubular phosphate alterations in patients with advanced CKD and bone mineral disorders, even in those with normal serum phosphate levels.

### 2.4. Uricosuria

In humans, uric acid is a weak acid with a pKa of 5.8, which exists as a urate at physiological pH, and is the final product of purine and ATP metabolism. Due to the absence of uricase, uric acid cannot be metabolized to allantoin, a water-soluble product that is easily eliminated by the kidney [70]. One-third of urate elimination occurs in the gastrointestinal tract and two-thirds occurs in the kidney [71], where it is freely filtered. Specific resorption occurs in the proximal tubule and is predominant in humans and rats compared to other species such as pigs and rabbits [72]. There are at least three transporters for reabsorption: urate transporter type 1 (URAT1; SLC22A12), OAT10 (SLC22A13) both in the apical membrane, and glucose transporter 9 (GLUT9; SLC2A9) in the basolateral membrane of the proximal tubule [73,74]. Uricosuric agents, such as losartan, probenecid, benzbromarone, and recently verinurad, are potent inhibitors of URAT1 [75,76,77].

A relevant aspect of renal urate handling is the 4-step model: glomerular filtration, proximal tubular reabsorption, tubular secretion, and post-secretory reabsorption [78]. Fractional urate excretion is approximately 10% in humans, significantly lower than in pigs, which is 200%) [79,80]. Urate is secreted by OAT1 (SLC22A6), OAT2 (SLC22A7), and OAT3 (SLC22A8) at the basolateral membrane of the proximal tubule, driven by the gradient created by the sodium/α-ketoglutarate cotransporter (SLC13A3). At the apical membrane, urate is secreted by MRP4, ABCG2, NPT1 (SLC17A1), and NPT4 (SLC17A3) [70].

Because uric acid is reabsorbed in the proximal tubule, urinary uric acid excretion is useful for functional evaluation of this portion of the nephron.

In an analysis by Kosmadakis et al., 44 patients with AKI were evaluated; 50% of cases were attributed to decreased renal perfusion, and the other half were attributed to other causes. None of the patients had used diuretics. A FEUA value of 23.79% had a sensitivity and specificity of 82% for the diagnosis of AKI secondary to decreased renal perfusion [81]. Berger et al. hypothesized that a fractional excretion of uric acid (FEUA) greater than 8% indicates a decongestive response when using loop diuretics, whereas a value lower than 5% is more likely in patients with acute kidney injury (AKI) due to volume depletion as an adverse effect of loop diuretic use [82].

Another analysis of 54 post-kidney transplant patients with AKI of undetermined etiology administered saline solution evaluated urinary tubular indices to predict improvement in volume status and renal function. In the logistic regression analysis, only FEUA was significantly for response to saline hydration (OR 0.92, 95% CI 0.85–0.99; *p* = 0.045), obtaining lower FEAU values than those who did not respond. FEUA was significantly lower among patients taking diuretics than non-users (10.2% vs. 15.4%, *p* < 0.05) [83].

It is well known that an important diagnostic clue that can differentiate between CWS (cerebral wasting syndrome) and SIAD (syndrome of inappropriate antidiuresis) is the depletion of volume, increased excretion of urate, and a recent history of cerebrovascular disease or traumatic brain injury [84,85]. In patients with CWS, increased FEUA can prevail even after correction of hyponatremia compared to SIAD [86]. The mechanism of uricosuria in CWS is related to reduced proximal tubule reabsorption accompanied by salt wasting [87]. Some important researchers have cast doubt on the existence of CSW [88].

Owing to the importance of the proximal tubule in uric acid reabsorption, nephrotoxic-induced cell injury can be assessed by quantifying uricosuria. In a prospective multicenter analysis in France evaluating 196 patients with hepatitis B virus infection divided into 3 groups: naïve, treated with entecavir (ETV), and treated with tenofovir (TDF), the subclinical proximal tubulopathy (SPT), defined by the presence of FEUA above 10% or TmP/eGFR below 0.8 mmol/L, was sought. The prevalence of SPT at 24 months of follow-up in the TDF group was higher than in the naïve group (50% vs. 30%; *p* > 0.05), although not statistically significant; but the HR: 2.8 for the cumulative incidence of SPT was significantly higher in patients treated with TDF vs. the naïve; the median survival time without SPT in TDF-group was 5.9 months [89]. These results are consistent with the findings of the Spanish MENTE study, in which uric acid excretion was significantly higher among patients treated with TDF vs. ETV (8.5% ± 4.2% vs. 6.9% ± 2.6%; *p* = 0.003) [89].

Although nearly 10% of the population experience hyperuricemia at least once in their lifetime, over 80% of them remain asymptomatic [90]. However, patients with CKD have an increased risk of hyperuricemia, and patients with hyperuricemia have an increased risk of CKD. However, currently, the treatment of asymptomatic hyperuricemia is not recommended to prevent the progression of CKD [91].

Asahina et al. in Japan studied 1042 patients with low eGFR (15–60 mL/min/1.73 m^2^), FEUA and urinary uric acid-to-creatinine ratio (UUCR) were calculated; median FEUA was 7.2%, and UUCR was 0.33 g/g. At a follow-up of 1.9 years, there were 30% composite renal events (50% reduction in eGFR compared to baseline or start of dialysis), the lowest quartile of FEUA exhibited an HR: 1.68 (95% CI, 1.13–2.5; *p* = 0.01); in fact, patients in the highest quartile of FEUA or UUCR had a lower risk of developing renal outcome [92]. This suggests that increased tubular urate reabsorption may be related to deleterious effects that favor the progression of kidney damage.

In another analysis performed in China of 625 patients with CKD (mean sCr 2.2 mg/dL, 13.7% CKD G4, and 15.5% CKD G5), a nonlinear relationship was observed between eGFR and urinary uric acid indicators. The FEUA, excretion of uric acid per volume of glomerular filtration (EurGF), clearance of uric acid (Cur), and glomerular filtration load of uric acid (FLur) were calculated. FEUA showed an inverse relationship with eGFR, with an overall value of 7.3% (5.5–11.3), but 11.1% (7.9–15.3) in patients with CKD G4 and 20.7% (15.6–29.7) in patients with CKD G5. Only Cur and FLur exhibited a linear association with kidney function in patients with eGFR, suggesting that these urinary indices are better parameters for monitoring uric acid metabolism in patients with CKD [93].

It is relevant to mention that in patients with gout and eGFR > 60 mL/min/1.73 m^2^, the urinary indices FLur, Cur, and FEUA were higher than in healthy subjects [94].

Even in T2D and obese patients, a relationship has been reported in which an increased FEUA was detected in both conditions, a clear mechanism associated with metabolic disturbances [95,96].

### 2.5. Citraturia

The plasma concentration was approximately 2.4 mg/dL. It forms complexes with sodium, calcium, and magnesium and binds very little bound to large molecules, so more than 90% of plasma citrate is freely filtered by the kidney. The unbound form of citrate is freely filtered by the glomerulus, and in humans, almost 75% of filtered citrate is reabsorbed [97,98].

Following glomerular filtration, the majority of citrate is reabsorbed in the proximal tubule, where it exists in equilibrium between its divalent and trivalent ionic forms, depending on the luminal pH. As proximal tubular cells actively secrete hydrogen ions and reabsorb bicarbonate, the Cit^2−^/Cit^3−^ ratio increases. The divalent form (Cit^2−^) is transported across the apical membrane via sodium-dicarboxylate cotransporter 1 (NaDC-1). Citrate metabolism generates HCO_3_^−^ ions that are transported across the basolateral membrane to the extracellular fluid [99].

In metabolic acidosis the amount of citrate excreted in urine decreases, and the amount of “new” bicarbonate generated by citrate metabolism increases [100,101].

Retention of acid (H^+^) seems to contribute to a gradual decrease in GFR in patients with CKD, including some without overt metabolic acidosis [102].

Due to the interest in subclinical metabolic acidosis and its potential role in the progression of chronic kidney disease, Goroya et al. hypothesized that urinary citrate may serve as an indirect marker of subclinical H^+^ retention. Following an oral sodium bicarbonate challenge, they measured the deviation between actual and expected plasma total CO_2_ levels. The study assessed the association between urinary citrate excretion and H^+^ retention in albuminuric CKD–G2 (n = 40) and those CKD–G3 (n = 26), before and after a 30-day intervention consisting of alkali-rich fruits and vegetables. As expected, patients with a lower glomerular filtration rate exhibited higher H^+^ retention and reduced citraturia. After the dietary intervention, there was evidence of decreased acid retention in the CKD–G2 group. These data are consistent with the relationship between decreased citraturia and hidden acid retention [103].

Gienella et al. conducted both a retrospective and a prospective cross-sectional study to investigate urinary citrate excretion across varying stages of kidney function. The retrospective analysis included 1733 individuals with a history of nephrolithiasis, where 24–h urinary citrate and the citrate–to–creatinine ratio were quantified. In the prospective arm, 22 participants with preserved renal function and 50 patients with different degrees of CKD were assessed using spot urine samples to determine the citrate-to-creatinine ratio. Notably, serum bicarbonate concentrations remained stable and within normal parameters until patients approached CKD stage 5. In contrast, urinary citrate excretion declined progressively and significantly starting from CKD stage 2. An important aspect of the study is that an inverse relationship was demonstrated between the spot citrate–to–creatinine ratio and net acid retention, a phenomenon not observed with serum bicarbonate [104]. These observations support the role of urinary citrate as a functional biomarker of early acid accumulation.

The alteration of tubular handling of solutes mentioned in this section clearly indicates a pathology involving the proximal tubule, without necessarily ruling out tubular pathologies in other segments of the nephron (Figure 2).

## 3. Non-Proximal Tubule Involvement

The term nonproximal tubular involvement describes dysfunction affecting the nephron segments beyond the proximal tubule. These include the thick ascending limb of the loop of Henle (TAL), distal convoluted tubule (DCT), and collecting duct (CD). Each segment critically contributes to solute handling and urine concentration, and its injury or dysfunction presents with distinct clinical and biochemical profiles.

### 3.1. Kaliuresis

Potassium is the most abundant cation in intracellular fluid and plays a key role in maintaining cell function. The human body has developed many mechanisms to control total K^+^ distribution [105]. The principal factors affecting the internal balance of K^+^ are insulin, catecholamines, plasma tonicity, and acid-base status. The kidney is the main organ responsible for maintaining the total body K^+^ content by harmonizing intake with excretion. Potassium was freely filtered through the glomerulus.

Of all the K+ filtered by the glomerulus, only 10% reaches the distal tubule, while most of its proximal resorption occurs alongside Na^+^ and water. In the thick ascending limb of the loop of Henle, the reabsorption process occurs through the Na^+^/K^+^/2Cl^−^ co-transporter (SLC12A1) [105].

Potassium secretion begins in the early distal convoluted tubule (DCT1) and progressively increases along the distal nephron into the cortical collecting duct. Under depleted conditions, H^+^/K^+^ ATPase reabsorption occurs in the collecting duct.

PURE was a prospective cohort that included 101,945 patients from five countries. They evaluated the correlation between hourly urinary sodium and potassium excretion and daily intake and found that the approach was less reliable for estimating potassium intake than sodium intake. However, it was also found that a higher estimated potassium excretion was associated with a lower risk of the composite of death and major cardiovascular events, possibly as a marker of healthy dietary patterns (high consumption of fruit and vegetables) [106]. 

A direct relationship has been observed between the usual intake of K^+^ in the diet with the measurement of 24 h urine K^+^, the higher the uK^+^, the higher the intake of healthy foods (vegetables, grains, fish, fruits). The corresponding urinary threshold values were 60 and 41 mmol/day in men and women, respectively. They also found that urinary excretion was inversely correlated with BMI (r = −0.149, *p* < 0.001), heart rate (r = −0.195, *p* < 0.01), and diastolic blood pressure (r = −0.139, *p* < 0.05) [107]. 

A similar finding was reported by the DASH group study, in which they observed an inverse correlation between higher levels of urine K^+^ excretion and blood pressure in 459 patients with arterial hypertension [108].

Kidney excretion declines during acute renal failure and chronic kidney disease. One of the main channels that are excreted in the distal nephron is the renal outer medullary potassium channel (ROMK). Rabb et al. found that the expression of this channel was reduced during AKI. They performed nephrectomy and 35 min of renal ischemia in Sprague-Dawley rats and measured the mRNA expression of ROMK in the outer medulla tissue. They found that the serum K^+^ level was increased at 24 h and further elevated at 48 h, concomitant with decreased urinary potassium. As expected, ROMK mRNA decreased by 70–80% at 48 h (*p* < 0.01) [109].

Gimelreich et al. obtained similar results in an analogous rat ischemia model. They performed one hour of ischemia by bilateral renal artery clamping and demonstrated downregulation of ROMK, which may explain hyperkalemia during AKI [110]. Another study conducted by Greevy et al. aimed to assess the contributions of distal tubular potassium secretory dysfunction and GFR reduction to the development of hyperkalemia in older patients. They were divided into three groups: older patients with hyperkalemia, older patients with normokalemia, and younger patients with normokalemia. All patients received a bolus of fludrocortisone and the transtubular potassium gradient (TTKG) was measured to assess distal tubular K^+^ secretion. They also measured renin, aldosterone, and GFR. There was no significant difference in the mean baseline TTKG between all three 3 groups but a significantly lower GFR was observed in the older patients. They concluded that TTKG was insufficient in older people with hyperkalemia [111].

Concerning hypokalemia, Li et al. evaluated the value of a spot urine test and 24-h urine potassium excretion in 67 patients for diagnosing hypokalemia caused by renal loss (use of diuretics, glucocorticoids, primary hyperaldosteronism, renal tubular acidosis, Bartter/Gitelman’s syndrome). On admission, a spot urine sample was collected, and a 24-h urine specimen was collected to calculate the urine K^+^/urine creatinine ratio (uK/UCr), spot sample potassium (uK), fractional excretion of potassium (FEK), and TTKG. In all patients, excretion was higher for all formulas. The cut-off for renal potassium losses in the 24 h uK was >42.27 mmol/24 h, and among all spot urine parameters, the strongest significant positive correlation was FEK (r = 0.83,1 *p* < 0.001) [112].

Elisaf 1995 studied 82 hypokalemic subjects and determined that a FEK of 15 ± 2 was related to renal potassium loss [113]. Yuan et al. evaluated the diagnostic value of spot FEK and 24-h FEK in patients with primary hyperaldosteronism compared with other urinary biomarkers. Spot FEK was more accurate for predicting renal potassium loss, with a cut-off of 9.8%, with sensitivity of 86% and a specificity of 87%, compared with a cut-off of FEK = 9.5% on a 24-h urine collection, with sensitivity = 80% and specificity = 65%. Interestingly, a uK/uCr = 5.5 mmol/mmol had a higher specificity (89%) but lower sensitivity (60%) [114].

Concerning managing hyperkalemia in predialysis, Caravaca-Fontán et al. conducted a transversal observational cohort study in 212 patients with CKD G4-5 (mean eGFR 15 mL/min/1.73 m^2^). They measured FEK and potassium charge relative to the glomerular filtration rate (uK/GFR). They found that in the study group, men had an inferior FEK compared to women, but with no significant differences in hyperkalemia between sexes, which may indicate better extrarenal potassium elimination in men than in women. Also, FEK showed a strong linear correlation with uK/GFR (r = 0.74), being significantly greater in patients with hyperkalemia compared with those with normokalemia (4.2 ± 1.5 vs. 3.7 ± 1.4 mmol/min/min, *p* = 0.04) [115].

### 3.2. Natriuresis

The control of sodium and water homeostasis by the kidneys has been the subject of multiple studies. The molecular description of the SLC12 family of electroneutral cotransporters was could until the molecular description of the principal transporters of electroneutral tubule [116,117]. Approximately 65% of all filtered sodium in the proximal tubule is reabsorbed because of its high epithelial ion and water permeability. Other relevant transporters in this segment are sodium-glucose cotransporters (SGLT), sodium-hydrogen exchangers (NHE3), and sodium-bicarbonate cotransporters (Na^+^/HCO_3_^−^, NBC). The NHE3 is quantitatively the most important, but other transporters are linked to other solutes that generally contribute to less than 5% Na^+^ reabsorption [116,118]. In the thick ascending loop of Henle, 25% of filtered sodium is reabsorbed through paracellular and active transport. The main transporter is NKCC2 (SLC12A2), which is the site of action of loop diuretics, such as furosemide and bumetanide. This segment is near the macula densa, a group of 15–20 specialized tubular epithelial cells, which sense tubular Na^+^ and Cl^−^ by NKCC2, NHE2, and Na^+^/K^+^ ATPase and regulate afferent arteriole vascular tone, affecting renal blood flow and filtration fraction [116,119]. In DCT, NCC (SLC12A3) reabsorbs approximately 5% of the filtered sodium, the target site of thiazide diuretics. in the cortical and medullary collecting duct (CD), about 1% of Na^+^ is reabsorbed. The principal cells drive Na^+^ reabsorption through the epithelial sodium channel (ENaC) and promote potassium secretion through ROMK. In intercalate cells, the chloride bicarbonate exchanger (Cl^−^/HCO_3_^−^ exchanger) or pendrin regulates Na^+^ reabsorption through altered ENaC activity by modulating bicarbonate or pH [116].

The importance of body sodium lies in its balance; therefore, the maximum amount of salt consumption recommended by the World Health Organization (WHO) is 5 g/day. High salt consumption increases blood pressure, which is the greatest risk factor for cardiovascular disease. The 24-h Na^+^ measurement is the recommended method for assessing sodium consumption in the population [120].

These recommendations were obtained from studies such as PURE, a multicenter study in which 101,945 patients were assessed with a mean follow-up of 3.7 years, and the association between estimated urinary sodium and the composite outcome of death and major cardiovascular events was examined. They compared the estimated sodium excretion of 4.00 to 5.99 g per day as a reference range; a value ≥7.00 g per day had an association with an increased risk of the composite outcome (OR: 1.15, 95% CI 1.02–1.30; *p* < 0.05).

Furthermore, the association between high estimated sodium excretion and the composite outcome was strongest among participants with hypertension (*p* = 0.02). Nevertheless, one of the limitations of this study was that sodium intake was estimated based on measured urinary excretion [106].

Although these formulas can underestimate high salt intake and overestimate low salt intake, they can be useful for assessing changes in 24-h uNa and monitoring changes in the intake pattern in different programs [120].

For many years, the greatest utility of measuring urinary sodium has been distinguishing between classical “pre-renal” and “intrinsic” AKI. Classically, uNa values < 20 mEq/L and a FENa < 1% indicated low perfusion damage, while a Na > 40 mEq/L and a FENa between 1–3% were indicators of acute tubular necrosis (ATN) [121]. It is important to consider that these studies included a few patients with considerably increased mean blood urea nitrogen and serum creatinine levels, suggesting that only patients with severe AKI were included [122]. It is currently known that this value is highly variable, with several confounding factors and often altered by fluid management, the use of vasoactive drugs and diuretics, having a FENa < 1% in circumstances of established AKI reflecting nonhomogeneous injury to the kidney parenchyma, and preservation of tubular function in these areas [123].

In patients with acute kidney injury, uNa may decrease as a result of a decrease in the glomerular filtration rate and avid sodium reabsorption; according to previous clinical and experimental studies, changes in urinary biochemistry precede serum changes in the development of AKI, suggesting a close relationship between urinary biochemistry and kidney function, a situation to be taken into account for periodic monitoring in spot samples of patients in whom we expect abrupt decreases in kidney function, such as postcardiac or kidney transplantation [123].

In addition, under these previous concepts, uNa may be useful in clinical situations where renal sodium avidity is a key physiopathology factor, such as AKI in cirrhotic patients. A Brazilian prospective cohort study included 225 adults to investigate uNa/uK and FENa levels as surrogate markers of sodium excretion in patients with decompensated cirrhosis. They found that the uNa/uK ratio was positively correlated with FENa and serum Na^+^ levels and negatively correlated with serum creatinine, total bilirubin, CRP, CLIF-SOFA, MELD, and Child-Pugh scores. The presence of ascites and Child-Pugh class C was associated with significantly lower values of sodium excretion parameters. Besides, AKI diagnosis at admission was associated with low uNa (20.0 mEq/L vs. 56.0 mEq/L; *p* < 0.001), uK (33.2 mEq/L vs. 38.8 mEq/L; *p* = 0.035) and uNa/uK (0.56 vs. 1.31; *p* = 0.003) compared to patients without AKI [124].

Another prospective analysis was performed in 200 patients with decompensated cirrhosis with AKI to evaluate the diagnostic utility of FENa for differentiating AKI phenotypes and to validate it in an independent cohort to differentiate ATN–AKI, hepatorenal (HRS)-AKI, and prerenal AKI. This group found that the median FENa level was significantly different in various phenotypes of AKI in the derivation and validation cohorts (*p* = 0.001). FENa cutoff of 0.56 showed an AUC of 0.86, a sensitivity of 89.4%, and specificity of 71.3% for differentiating ATN–AKI from non-ATN–AKI (*p* = 0.001). More interestingly, a cut-off value of 0.44 showed an AUC of 0.74 with a sensitivity of 77.9% and specificity of 65.9% for differentiating HRS–AKI vs. non-HRS–AKI (*p* < 0.001) [125].

An important finding in patients using loop diuretics for evaluating AKI is that the drug can influence natriuresis; therefore, the fractional excretion of urea (FEUrea) may be helpful in patients receiving diuretics [126].

However, FEUrea may lack specificity in evaluating patients with AKI, and FENa remains the best marker for differentiating between transient and persistent AKI [127].

A recent meta-analysis involving 11 studies and 1108 hospitalized patients similarly indicated that FEUrea has limited utility in distinguishing between intrinsic and prerenal AKI, even among patients receiving diuretics [128].

Another group of patients in whom the measurement of uNa has obtained promising results is those with heart failure, where activation of compensatory mechanisms results in low renal perfusion, hypoxia of tubular cells, and damage. Recent studies have shown that uNa measurement has prognostic value associated with diuretic treatment in these patients. Furosemide stress tests (FST) were used to evaluate renal tubule integrity in AKI. The test can predict the risk for severity progression with a bolus of furosemide (1 mg/Kg for naïve or 1.5 mg/kg for pre-exposed patients), with an AUC = 0.68 sensitivity of 87.1% and specificity of 84.1% [129,130,131]. In fact, in an interesting study, 84 subjects were evaluated to determine the FST response in patients with fibrosis, and as expected, compared with a low degree of fibrosis, patients with the highest degree of fibrosis showed the lowest urine output at the first hour (*p* = 0.015) [132].

A group of patients in whom the measurement of uNa has obtained promising results is those with heart failure, in which the activation of compensatory mechanisms results in low renal perfusion, hypoxia of the tubular cells, and damage. In these patients, the uNa measurement has a prognostic value associated with diuretic treatment.

In the Renal Optimization Strategies Evaluation (ROSE)-AHF study, a uNa ≤ 60 mmol/L within the first 24 h was associated with an increased risk of longer hospitalization. Early monitoring is suggested, which consists of achieving a uNa of 50–70 mmol/L 2 h after loop diuretics and/or a urinary output of 100–150 mL/h. Six hours after treatment, the patient showed an inadequate response to diuretics. Therefore, uNa in patients with acute heart failure is related to their clinical status and has prognostic value between the early phase of hospitalization and discharge [133].

### 3.3. Calciuria

Calcium is fundamentally involved in cell signaling via direct transduction, or as a secondary messenger. Daily Ca^2+^ absorption occurs through three main mechanisms: bone turnover, intestinal absorption, and renal reabsorption. Although they could be considered equivalent mechanisms, the kidney has been identified as the main actor in calcium metabolism [134,135,136]. The regulation of Ca^2+^ homeostasis is influenced by PTH, calcitriol (1,25-dihydroxy vitamin D3), calcitonin, fibroblast growth factor 23, and the serum Ca^2+^ concentration itself [135]. In plasma, approximately 50% of calcium is found free (ionized, iCa). In contrast, the other half is found in complexes bound to proteins (albumin) or other organic anions (citrate, oxalate, bicarbonate) [137]. It is considered that about 10 g of calcium are filtered in the glomerulus per day, and about 100 to 200 mg of calcium are excreted per day [138]. The proximal convoluted tubule is responsible for the reabsorption of 60% to 70%, the loop of Henle 20%, the distal convoluted tubule 10%, and finally, the collecting tubule only 5% of the filtered Ca^2+^ [135].

Transport in the convoluted proximal tubule is essentially passive; most transport is mediated by paracellular claudin-like transporters in subtypes 2, 10a, and 17 [135]. In the Pars recta (S3), L-type voltage-dependent Ca^2+^ channels and transient potential receptor channels are relevant. CaSR can be found in the extension of the proximal tubule, which when activated, causes a change in the expression of vitamin D receptors [136].

According to Hodgkinson’s study, in healthy subjects, the range of calciuria in 24 h is 100–300 mg/day [139]. Calciuria can be influenced by modifiable factors, such as weight and 25-O-Hydroxyvitamin D3 intake, and non-modifiable factors, such as ethnicity and menopause. The consumption of 500 mg of calcium can increase urinary excretion by approximately 25 mg [140].

In a study performed on 317 healthy controls, 24-h calciuria was measured, and the mean of uCa/uCr was 0.4 ± 0.24 mmol/mmol, higher in non-menopause women compared with males (0.43 ± 0.25 vs. 0.27 ± 0.16). The ratio was significantly higher in menopausal women (0.45 ± 0.26) [141].

Cameli et al. evaluated 237 patients with sarcoidosis and compared them with 40 patients with interstitial pulmonary fibrosis and 28 patients with chronic hypersensitivity pneumonitis. They found that subjects with sarcoidosis had significantly higher levels of serum and urinary Ca^2+^ (*p* < 0.001); the former showed a better diagnostic accuracy than serum Ca^2+^ in discriminating sarcoidosis from non-sarcoid lung fibrosis (AUC 0.7658 vs. 0.6205; *p* = 0.0026 vs. *p* = 0.1820) [142].

Toxic metals can induce tubulopathies, and Cd has been shown to induce tubular cell dysfunction. Xu et al. compared 499 subjects exposed to a cadmium-polluted area with 252 control subjects. Urinary cadmium and calcium levels were higher in the cadmium-exposed group than in the non-exposed group. There is a relevant dose-response relationship between urinary cadmium and hypercalciuria; tubular dysfunction is also related to higher levels of urinary beta 2-microglobulin and N-acetyl-bet-D-glucosaminidase [143].

Gitelman and Bartter syndromes are difficult to separate from biochemical data alone, especially when overlapping electrolyte and acid-base disturbances occur. Gitelman syndrome and Type 3 Bartter syndrome can have identical biochemical parameters. In the past, the thiazide protocol was used to guide differential diagnosis, but it is currently not recommended [144]. Therefore, calciuria is the most sensitive marker for differentiating between these two entities. Patients with Bartter syndrome [145].

Hagras et al. conducted a study on 246 pregnant women after 20 weeks of gestation, evaluated uCa/uCr, and followed up until delivery. Compared with normotensive women, a uCa/uCr ≤ 0.04 had 79.3% sensitivity, 96.3% specificity, 91.5% positive predictive value (PPV), and 90.3% negative predictive value (NPV) in the prediction of preeclampsia [146]; similar findings were detected by other researchers [147].

Using calciuria assessment methods, such as spot uCa/uCr, would be physiologically consistent in predicting the response to hyperhydration treatment in patients with severe hypercalcemia.

Hypercalciuria has been recognized as an important risk factor for nephrolithiasis; most kidney stones are composed of calcium [144,148]. The most recognized pathological mechanism of stone formation is supersaturation, which is related to fluid intake and urinary calcium levels. For example, according to a study conducted by Borghi et al., patients with lower water intake (<2 L/day) had more recurrences during a 5-year follow-up (*p* = 0.008) [149]. A urinary calcium level > 200 mg/day has been associated with an increased risk of nephrolithiasis [150]; some studies have suggested even a lower threshold of 150 mg/dL [151], even though it has not been widely tested in this specific setting. A uCa/uCr > 0.11 in fasting individuals and >0.22 after calcium intake has been related to hypercalciuria [152].

Given its association with stone formation, some medications have been studied for their hypocalciuric properties, such as thiazides. A volume contraction secondary to the use of these drugs leads to a compensatory increase in sodium reabsorption from the proximal tubule, which in turn causes increased calcium reabsorption, thereby reducing urinary calcium levels [148]. In a meta-analysis of which 511 patients were included, the incidence of nephrolithiasis was significantly lower in the thiazide-use group (95% CI 0.33–0.58; *p* < 0.0001), in addition to lower urinary calcium levels. However, thiazide use was associated with a higher incidence of adverse effects [153].

A randomized placebo-controlled clinical trial (NOSTONE trial) on 416 patients with urolithiasis did not find advantages with the prescription of thiazides at three different doses (12.5, 25, and 50 mg), and there was no relation between the dose and lithiasis recurrence (*p* = 0.66) with more adverse events related to thiazide (hypokalemia, gout, diabetes, and rising serum creatinine) compared to placebo [154]. The trial systematically excluded patients with CKD, women, and those with secondary causes of nephrolithiasis. Moreover, previous positive trials used a dose of 50 mg or higher of hydrochlorothiazide (only 35% of patients in the NOSTONE trial received 50 mg). Similarly, indapamide and chlorthalidone seem to be more potent and potentially have more biological effects.

### 3.4. Magnesuria

Magnesium (Mg^2+^) is the fourth most common element on Earth and the second most abundant intracellular cation in the human body. Mg^2+^ is involved in major cellular and physiological processes, primarily through its nucleotide-binding properties and regulation of enzymatic activity. Interestingly, Mg^2+^ has two hydration shells, resulting in a hydrated radius nearly 400 times larger than its dehydrated radius. Therefore, Mg^2+^ must undergo dehydration to be transported through the channels [154,155,156,157]. Mg^2+^ absorption through enterocytes into the systemic circulation occurs through an active saturable pathway (transcellular) and a passive non-saturable pathway (paracellular). Uptake occurs mainly through the paracellular pathway in the distal portions of the jejunum and ileum. The final modulation of Mg^2+^ absorption occurs in the cecum and colon mediated by the membrane transient receptor potential melastatin type 6 and 7 (TRPM6/7) proteins. It has been proposed that the mechanism of basolateral transport of Mg^2+^ from enterocytes is through Cyclin M4 (CNNM4), a Na^+^/Mg^2+^ antiporter [158].

The Mg^2+^ reabsorption in the early proximal tubule is virtually absent [159]; reabsorption (15–20%) depends on paracellular permeability by claudin-2 and -12 in the late proximal tubule [160]. Increased serum Mg^2+^ has been consistently observed in large clinical trials with SGLT-2 inhibitors [161,162]. Due to the limitation of Na^+^ absorption in the proximal tubule, its concentrations in the distal tubule increase, potentially favoring the reabsorption of Mg^2+^, particularly in the thick ascending limb [160]. The main site of Mg^2+^ reabsorption along the nephron is the TAL (60–65%). In this segment, the claudin-16/19 complex provides a cation-selective pore for paracellular reabsorption, regulated by CaSR, PTH, and mechanistic target of rapamycin (mTOR) signaling [155,160,163]. Patients with chronic use of loop diuretics that block NKCC2 develop renal Mg^2+^ loss as it decreases the generation of transepithelial voltage necessary for paracellular reabsorption [164]. Fine changes in Mg^2+^ reabsorption (5–15%) occur in the DCT, where transcellular transport occurs through TRPM6/TRPM7 channels. The basolateral expulsion of Mg^2+^ has been proposed to be dependent on the Na^+^ gradient. Cyclin M2 (CNNM2) and Solute Carrier Family 41 member 3 (SLC41A3) are the main candidates that act as Na^+^/Mg^2+^ exchangers; disturbances in this exchange indirectly result in impaired renal Mg^2+^ reabsorption in the DCT [163].

Elisaf et al. found that in patients with hypomagnesemia and normal renal function, the fractional excretion of magnesium (FEMg) could better distinguish between renal and no renal loss of Mg^2+^ than the uMg/uCr ratio, and showed that a value >4% is indicative of significant renal loss [165].

Futrakul et al., In a very interesting study, evaluated 129 patients with nephrotic syndrome with various degrees of tubulointerstitial damage and demonstrated that in patients with more severe tubulointerstitial fibrosis (score 69 ± 19%), all fractional excretions of solutes (Na^+^, Ca^2+^, P^−^, uric acid, and Mg^2+^) were significantly different. However, FEMg was the most sensitive index for detecting early tubular dysfunction (r = 0.88, *p* < 0.001) [166].

Gheissari et al. evaluated 20 children with a history of clinically recovered acute ischemic tubular necrosis. FEMg was significantly higher in the study group than in the control group, suggesting that FEMg can be used to detect early stage chronic kidney disease [167].

Deekajorndech et al. found high FEMg in kidney diseases associated with tubulointerstitial fibrosis, such as focal segmental glomerulosclerosis and CKD. Both normoalbuminuric and microalbuminuric patients with type 2 diabetes had elevated FEMg levels. Multiple regression analysis showed an inverse correlation between FEMg and peritubular capillary flow [168]. Futrakul et al. proposed increased FEMg as a biomarker that can recognize early diabetic kidney disease in conjunction with decreased creatinine clearance, glomerular filtration rate, and decreased cystatin C [169].

Several drugs are associated with the development of hypomagnesemia. Proton pump inhibitors are among the most important causes of hypomagnesemia. They induce this condition by affecting the luminal pH and intestinal microbiota and by causing defects in the absorption of Mg^2+^ mediated by the TRPM6/7 block [170]. Four different renal mechanisms have been described in drug-induced hypomagnesemia: decreased activity of TRPM6 and TRPM7 in the distal convoluted tubule (e.g., anti-EGFR and calcineurin inhibitors), reduction of paracellular transport in the thick ascending loop of Henle (diuretics, aminoglycosides, and mTOR inhibitors), alteration of the structural integrity of the distal convoluted tubule (thiazide diuretics, cisplatin, and amphotericin), and Mg^2+^ extrusion driven by Na^+^/K^+^-ATPase (digoxin) [171].

Interestingly, tacrolimus, an immunosuppressant widely prescribed for kidney transplantation and glomerular diseases, binds to FKBP12, an immunophilin, to inhibit the calcineurin protein and exert its immunosuppressive effects. Research has shown that tacrolimus induces a decrease in the mRNA expression of NCX-1, Calbindin 28k, and TRPM6; however, KS-FKBP12-/- mice treated with tacrolimus were protected from these effects [172].

Hypomagnesemia in critically ill patients increased the probability of mortality (RR, 1.9; 95% CI: 1.48–2.44; *p* < 0.001; I^2^ = 63. 5%) [173]. The magnesium depletion score (MDS) is a comprehensive scoring tool used to evaluate the status of magnesium deficiency. A high MDS suggests greater severity of magnesium deficiency. A recent study that included 5011 patients found that a high magnesium depletion score was associated with a significantly higher risk of all-cause and cardiovascular deaths among patients with previous cardiovascular disease [174].

### 3.5. Urine Osmolality

The mechanism of urinary concentration is one of the primary functions of the kidney. It is characterized by the formation of an osmolar gradient that increases along the medulla. This osmotic gradient is formed by the accumulation of solutes, particularly NaCl and urea, in cells, interstitium, tubules, and vessels of the medulla [175].

The mechanisms for independent control of water and sodium are mostly contained within the renal medulla. The medullary nephron segments and vasa recta are arranged in complex and specific anatomic relationships with respect to the three-dimensional configuration [176]. The pioneering studies in describing the mechanism of countercurrent multiplication hypothesized that the net transport of solutes and water, accompanied by the countercurrent capillary flow through the vasa recta, is what generates an osmotic difference between the tubular flow and the interstitium that allows urinary concentration. Vasa recta achieves osmotic equilibration through water absorption and solute secretion, as they are freely permeable to water, urea, and sodium to prevent dissipation of the osmolar gradient (countercurrent exchange) [177,178,179]. Other theories have emerged, such as the peristaltic contractions of the pelvic wall and the inner medullary lactate accumulation, but these have not been proven [180,181].

Changes in urinary osmolality can be very useful in many circumstances, such as the progression of CKD, response to treatment with tolvaptan in autosomal dominant polycystic kidney disease (ADPKD), and progression and mortality from AKI in critically ill patients.

In a prospective observational study by Tabibzadeh et al., fasting urine osmolality was associated with CKD progression, GFR decline, and pre-ESKD mortality. Adjusted HRs for ESKD were significantly higher in patients with lower baseline fasting urinary osmolality (lowest vs. highest tertile: aHR, 1.97; 95% IC 1.26–3.08; *p* < 0.05), patients with urine osmolality in the lowest tertile had a steeper mGFR per year decline compared to the highest tertile (−4.9 ± 0.9%, IQR −6.6% to −3.2%; *p* < 0.001) [182]. Hebert et al. also examined the prognostic value of urine osmolality tertiles on CKD progression; the crude HRs were significantly higher among patients in the lowest tertile (vs. higher tertile: aHR 10.8, 95% CI 7.47–15.82; vs. middle tertile: aHR 5.95, 95% CI 4.04–8.76) [183]. See Figure 3.

Uosm has also been studied in the context of ADPKD progression. In the post-hoc analysis from “TEMPO 3:4 trial”, 1037 patients were included. The baseline Uosm was 504 ± 177 mOsm/kg and was negatively correlated with age (r^2^ = 0.041, *p* < 0.001) and total kidney volume (TKV) (r^2^ = 0.044, *p* < 0.001), and positively correlated with eGFR (r^2^ = 0.117, *p* < 0.001). In the multivariate analysis, the baseline Uosm in the lowest quartile had a negative correlation with hypertension and TKV and a positive correlation with eGFR (Uosm increased 1.004 times or every 1 mL/min/1.73 m^2^ increase in baseline eGFR). As expected, the initial decline in uOsm in response to tolvaptan was positive correlated with the baseline Uosm (*p* = 0.003, r^2^ = 0.24) and eGFR (*p* < 0.001, r^2^ = 0.06).

The relationship between baseline Uosm and eGFR outcomes was evident, so the higher the baseline Uosm there was less deterioration of eGFRe (*p* = 0.002) (152).

Akihisa et al. evaluated the changes in Uosm after tolvaptan prescription and clarified the relationship between these changes and AQP2. 72 patients were included; Uosm was measured five times, and the urine aquaporin-2 to creatinine ratio was determined. The mean Uosm immediately before tolvaptan initiation was 351.8 ± 142.2 mOsm/kg, which decreased to 97.6 ± 23.8 mOsm/kg in the evening. Although Uosm increased 1 month later (mean 160 mOsm/kg), it was still significantly lower than that before tolvaptan initiation. The initial Uosm drop was negatively correlated with the annual eGFR change (β = −0.29, *p* = 0.14) [184].

Therefore, it is not surprising that polyuric syndromes should be evaluated initially with uOsm to determine whether the urine volume depends on the lack of arginine vasopressin (secretion or central production) or water diuresis by other causes (e.g., hypokalemia, hypercalcemia) (uOsm < 150 mOsm/L), or if it is related to an excess of solutes accompanied by osmotic diuresis (uOsm > 300 mOsm/L), or mixed (150–300 mOsm/L) [185].

### 3.6. Urine Anion GAP/Osmolal GAP

The urine anion GAP (UAG) is the difference between the principal cations (sodium and potassium) and principal anions (chloride): (uNa + uK − uCl). Typically, the sum of sodium and potassium absorbed from the diet in the gastrointestinal tract exceeds the absorbed chloride; therefore, the normal UAG is positive (20–100) [186]. In gastrointestinal losses such as acute diarrhea, cations losses exceed chloride, and the UAG becomes negative (<20) in the context of normal anion GAP metabolic acidosis (NAGMA) [187,188]. These cations are supplied by ammonium in urine along with chloride (NH4Cl) to maintain electroneutrality [186]. However, in patients with a reduced capacity to excrete ammonium, as in distal renal tubular acidosis, with NAGMA but without gastrointestinal loss of cations, their UAG reflects the diet (more cations absorbed than chloride), and the UAG is positive. In addition, it is important to consider that chronic respiratory alkalosis promotes reduced ammonium excretion and UAG can be positive [189]. In cases of suspected ethylenglicol intoxication, urinary osmolal GAP (UOG) is more useful.

An attractive and easy way to estimate urine ammonium is the urine osmolal gap (UOG), especially when excreted with anions, such as chloride [190,191]. The normal value of UOG is between 10 and 100 mOsm/L; if half of that excretion is ammonium, its concentration would be 5–50 mEq/L. The ammonium surrogate should be (measured UAG − calculated UAG)/2 [192,193,194]. It is important to consider that it represents only an approximation of the real ammonium concentration and should not be used as an exact quantitative measure of the ammonium concentration.

In contrast to UAG, UOG is not altered by a high concentration of unmeasured anions in urine.

The principal use is related to the evaluation of metabolic acidosis. In patients with renal tubular acidosis type 1 (distal RTA), the UOG level is usually below 150 mOsm/L. Patients with hyporeninemic hypoaldosteronism (type 4 RTA), typically those with mild to moderate deterioration of GFR and type 2 diabetes, can also have low urinary ammonium excretion [192,193]. Conversely, when ammonium excretion exceeds 200 mEq/L, the UOG is approximately 400 mOsm/L, as observed in patients with toluene intoxication and chronic diarrhea [193].

Of course, there are limitations to using this surrogate for ammonium excretion, such as urinary tract infections caused by urease-producing bacteria.

### 3.7. Urine pH

Urinary pH is a critical parameter that reflects the acidity or alkalinity of urine and, indirectly, renal control of serum pH. It is influenced by diet, renal function, and metabolic processes. The kidney is critical in regulating blood acidity by performing key functions: HCO_3_^−^ freely filtered through the glomeruli is almost entirely reabsorbed in the tubular system and generating “de-novo” HCO_3_^−^. When the kidneys excrete acid, it effectively produces a base or bicarbonate (HCO_3_^−^) [195,196]. The nephron’s ability to eliminate free protons (H^+^) is restricted, with a urinary H^+^ concentration of less than 0.1 mmol/L even at a urine pH of 4.5. Acid excretion primarily occurs in the distal nephron through two mechanisms: discharge of titratable acids and elimination of ammonium (NH_4_^+^). Therefore, the net acid excretion (NAE) in urine is computed by adding these components and subtracting urinary HCO_3_^−^, which is minimal under typical fasting conditions [197].

Non-anion-gap metabolic acidosis can arise from bicarbonate loss or reduced renal acid excretion. In healthy individuals, metabolic acidosis typically results in a urinary pH ≤ 5.3 or lower. However, exceptions occur in chronic metabolic acidosis under conditions such as hypokalemia, where renal cells responding to intracellular acidosis secrete H^+^ and ammonia (NH_3_), maintaining a urinary pH of 5.5 or higher [198].

Renal tubular acidosis (RTA) is an important condition. It can be differentiated based on urinary pH. A urinary pH below 5.5 generally excludes distal RTA but not proximal RTA. Proximal RTA is characterized by the impaired reabsorption of filtered bicarbonate in the proximal tubule. When serum bicarbonate levels are low, most filtered bicarbonate is reabsorbed, allowing for normal distal acidification and adjustment of urine pH according to dietary needs. After exposure to acids, urine pH typically decreases to 5.3 or lower [198,199,200].

In non-anion gap acidosis, a urinary pH of 5.5 or higher suggests distal RTA, where there is inadequate excretion of daily acid load, leading to hydrogen ion retention. Diagnostic criteria included persistent urine pH > 5.5, urine sodium concentration > 25 mmol/L, and reduced urine ammonium excretion. In incomplete distal RTA, impaired urine acidification prevents pH reduction below 5.3 with acid load, despite adequate net acid excretion. Patients with incomplete distal RTA have reduced urine citrate levels, which have been associated with nephrolithiasis, nephrocalcinosis, and osteoporosis [199].

Another condition that has gained relevance in terms of urine pH is metabolic syndrome, an interconnected syndrome that substantially elevates the risk of developing heart disease, stroke, and type 2 diabetes. Initially, research linked metabolic syndrome to urine pH levels in patients with uric acid nephrolithiasis. Studies have revealed a lower urinary pH with a higher prevalence of uric acid stones among obese individuals and those exhibiting metabolic syndrome traits [201,202].

The parameters associated with low urinary pH in metabolic syndrome have been established. Environmental risk factors such as smoking, alcohol intake, and a low-fiber diet are known contributors in the Asian population; however, in the Western population, dietary factors alone cannot account for more acidic urine [203,204]. The urinary pH is lower in patients with type 2 diabetes, hypertriglyceridemia, and a high waist-to-hip ratio. An inverse correlation between urinary pH and several metabolic syndrome component factors has been established [203,204,205].

The reason for the relationship between metabolic syndrome and urinary acidity remains unclear. Low urine pH could result from heightened excretion of net acids and compromised buffering owing to inadequate ammonium excretion in the urine.

It has been hypothesized that the mechanisms of acid-base homeostasis may be altered in a state of insulin resistance resulting from metabolic syndrome [206].

Even in the general population, during a 10-year follow-up period of 12,476 participants, it was found that 6.5% of men and 3% of women developed type 2 diabetes, with a cumulative incidence of 7.5% per 100 person-years. Men had a greater risk if urine pH ≤ 5.0 or 5.5 compared with urine pH ≥ 6.5 (HR: 1.93 and 1.46, respectively) [207].

In other contexts, the impact of low urinary pH has been studied across various contexts of AKI, including its role in surgical settings, critical illness, prevention of contrast nephropathy, and rhabdomyolysis-induced AKI. While low urinary pH has been linked to increased AKI incidence in surgical and traumatic rhabdomyolysis scenarios, further research has not consistently associated it with AKI progression [208]. Trials investigating urinary alkalinization with sodium bicarbonate aimed to protect the kidneys by potentially mitigating free radical formation under acidic conditions. However, evidence supporting the efficacy of sodium bicarbonate infusion is lacking and its use may be associated with higher vasopressor and fluid requirements. Notably, alkaline urine did not prevent AKI progression [208,209,210,211].

### 3.8. Urine Sodium to Potassium Ratio

Aldosterone regulates blood pressure and the exchange of sodium and potassium in the late distal convoluted tubule (DCT2), connecting tubule (CNT), and collecting duct. The reabsorption of Na^+^ by ENaC promotes an electrochemical gradient that favors potassium exit from the cell, as performed by ROMK. It is intuitive to use uNa/uK to evaluate aldosterone activity.

Segawa et al. evaluated second-morning urine samples from 160 hypertensive patients with eGFR > 45 mL/min/1.73 m^2^. Urinary Na^+^ and uNa/uK were lower and plasma renin activity (PRA) was higher in the high plasma aldosterone concentration (PAC) group. At the same time, the serum K^+^ levels were lower in the high-PAC group. The ROC curve between uNa/uK and high PAC was 0.77 (95% IC: 0.59–0.95) and 0.64 (95% IC: 0.36. 0.93) in men and women, respectively. A cutoff point for uNa/uK < 1 had a high specificity of 97.9% but a low sensitivity of 45.5% for diagnosing hyperaldosteronism [212].

In the INTERSALT study, the correlation coefficient between spot uNa/uK and 24-h Na/K ratio was r = 0.96 in analyses across populations (n= 52) and r = 0.69 in analyses across individuals (n= 10,065), with a bias of approximately 0.4, which is useful when 24-h collected urine is difficult to perform in daily clinical practice [213]. The same researchers performed another analysis on a subgroup of patients with hypertension; the mean uNa/uK ratio of random casual urine samples on four or more days was strongly correlated with the uNa/uK ratio of 7-day 24-h urine (r = 0.80–0.87), a similar correlation between 24-h urine from 1 and 2 days and 24-h urine from 7 days (r = 0.75–0.89) [214].

In the acute setting, the uNa/uK ratio obtained 48 h after admission was evaluated in 225 patients with cirrhosis. AKI was observed in 32% of patients and was associated with a lower uNa/uK ratio. AKI was diagnosed in 44% of patients with an uNa/uK ratio < 1, but only 8% of patients with a ratio of ≥2 [124]. This makes sense because a very low ratio reflects severe secondary hyperaldosteronism and, perhaps, a reduced probability of responding to diuretics.

Contrary to this assumption in patients with cirrhosis, subjects who consume too much sodium and too few fruits and vegetables tend to have a high uNa/uK ratio in the general population. Dietary interventions focused on lowering blood pressure could reduce the uNa/uK ratio [215]. Previous reports have shown associations between a higher uNa/uK ratio and increased risks of hypertension, cardiovascular disease, and mortality [216,217,218]. For instance, a low uNa/uK ratio is favorable in this context.

In another interesting study, Takase et al. evaluated 14,549 subjects without CKD criteria and measured the uNa/uK ratio using an overnight urine sample. At 61 months of follow-up, 25.9 per 1000 person-years developed CKD. The risk was more patent in the highest quartiles of the baseline uNa/uK ratio, being relevant in the multivariate analysis, where the ratio was a predictor of new-onset CKD after adjustment (HR: 2.0, 95% IC: 1.66–2.44; *p* < 0.001) [219]. Even in patients with previously diagnosed CKD, a higher uNa/uK ratio was associated with poor outcomes, including end-stage kidney disease [220].

Similar findings were reported in the Tromso substudy (RENIS-T6), which included 1311 subjects with a mean uNa/uK of 1.43 ± 0.76. A 1-standard deviation increase in the uNa/uK ratio was associated with increased systolic ambulatory blood pressure by 1 mmHg, not mediated through renal function variables such as measured GFR, urine albumin-urine creatinine ratio, and urine epidermal growth factor/creatinine ratio [221].

Table 1 summarizes the clinical relevance of the functional markers.

## 4. Conclusions

Adequate evaluation of renal function is essential in decision making to establish a diagnosis, treatment, and prognosis for patients. Renal function is usually glomerulocentric; however, the tubular system is indispensable for adequate homeostasis. Understanding how to assess tubular function and its clinical implications can enhance treatment options and influence the prognosis of common kidney and other systemic diseases.

## Figures and Tables

**Figure 1 pathophysiology-32-00033-f001:**
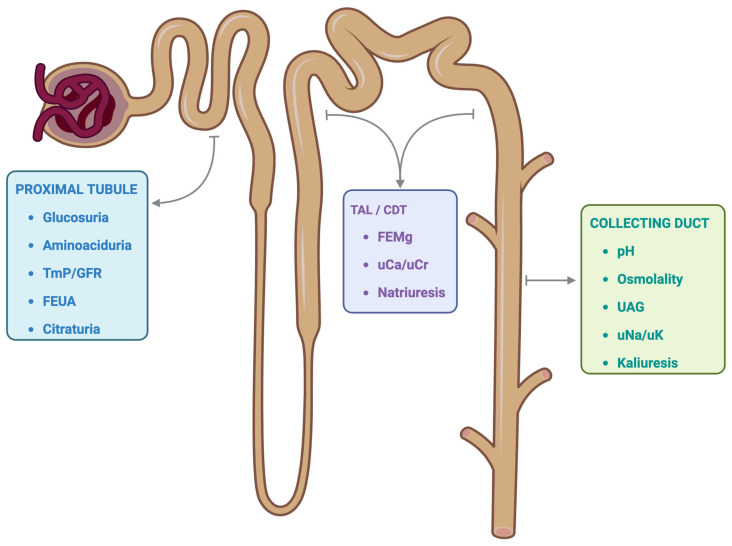
Tubular segment and potential functional markers. The functional markers of the renal tubular system are represented according to the segment where the most relevant regulation of each solute occurs. However, it is essential to note that various functional markers may depend on alterations in other anatomical segments of the tubular system. TmP/GFR: ratio of tubular maximum phosphate reabsorption to glomerular filtration rate. FEUA: Fractional excretion of uric acid. TAL: Thick ascending limb. CDT: Convoluted distal tubule. FEMg: Fractional excretion of magnesium. uCa/uCr: urine calcium to urine creatinine ratio. UAG: Urine anion GAP. uNa/uK: urine sodium to urine potassium ratio. Created with BioRender.com.

**Figure 2 pathophysiology-32-00033-f002:**
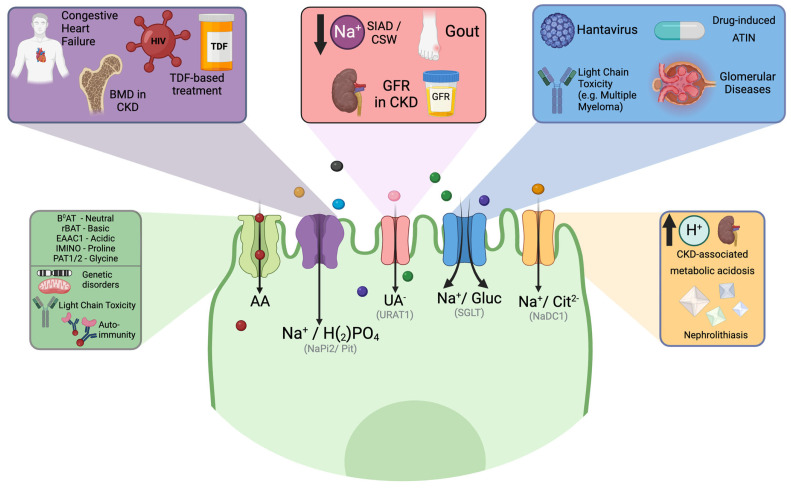
Clinical relationships involving the proximal tubule and potential transporters involved. Proximal tubule injury may be linked to solutes filtered by the glomerulus, drugs, or viruses that directly or indirectly affect the functioning of specific transporters. Additionally, a metabolic disorder can also result from functional defects in some specific transporters of the proximal tubule. ATIN Acute tubulointerstitial nephritis, BMD Bone mineral disease, BOAT1 Sodium dependent neutral amino acid transporter, CKD chronic kidney disease, CSW Cerebral wasting syndrome, EAAC1 Excitatory amino acid transporter 3, GFR Glomerular filtration rate, IMINO Signaling threshold-regulating transmembrane adapter 1, NaDC–1 Secondary active sodium dicarboxylate cotransporter–1, PAT1 Co–transporter of glycine–hydrogen, rBAT Neutral and basic amino acid transport protein, SGLT Sodium-glucose co-transporters, SIAD Syndrome of inappropriate antidiuresis, TDF Tenofovir, Na^+^/H_2_PO_4_ Sodium-dependent phosphate transport protein 2b, URAT1 Solute carrier family 22. Created with BioRender.com.

**Figure 3 pathophysiology-32-00033-f003:**
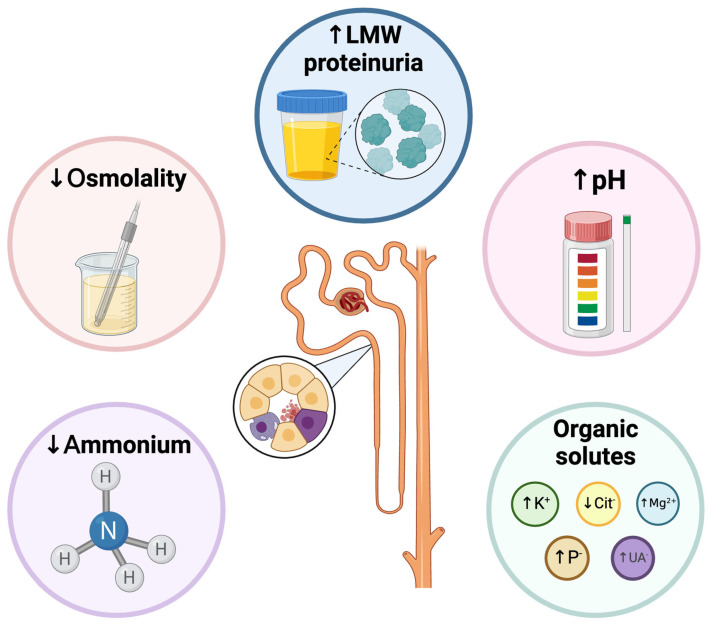
Functional alterations that are usually detected in acute or chronic tubular injury. Functional impairment may occur concomitantly with impairment of glomerular function or in isolation. LMW: Low to medium weight. Created with BioRender.com.

**Table 1 pathophysiology-32-00033-t001:** Functional markers and the method of measure.

Solute	Functional Marker	Method	Clinical Evidence	Ref.
Glucose	Glucosuria	Dipstick 24-h collection	ATIN Glomerulopathies	[17,18,19,20,21,22,23,24,25,26]
Amino acids	Aminoaciduria	Urine collection	Monoclonal gammopathies	[49,50,51,52,53,54]
			Diabetes, AKI	
Phosphate	FEP	uP × sCr sP × uCr	HF, HIV, pre-donation, SPT, CKD	[66,67,68,69]
	TRP	(1 − FEP) × 100		
	TmP/GFR	(α × sP) Where α = [0.3 × TRP]/1 − (0.8 × TRP)		
Uric Acid	FEUA	uUA × sCr sUA × uCr	Gout, AKI, hyponatremia, SPT, CKD	[82,83,84,85,86,87,88,89,90,91,92,93,94,95,96]
	Total uricosuria	24-h collection		
Citrate	Total Citraturia	24-h collection	SMA, Metabolic, syndrome, Lithiasis, CKD	[102,103,104]
	uCit/uCr	uUA × sCr sUA × uCr		
Potassium	Total kaliuresis	24-h collection	Diet, HBP, AKI, Inherited tubulopathies	[106,107,108,109,110,111,112,113,114,115]
	uK/uCr	uPotassium uCreatinine		
	FEK	uK × sCr sK × uCr		
Sodium	Total natriuresis	24-h collection	Diet AKI HF Diuretic monitorization	[106,120,121,122,123,124,125,126,127,128,129,130,131,132,133]
	FENa	uNa × sCr sNa × uCr		
	uNa	Spot urine sodium		
Calcium	Total calciuria	24-h collection	Toxics Preclampsia Lithiasis	[139,140,141,142,143,144,145,146,147,148,149,150,151,152,153,154]
	uCa/uCr	uCalcium uCreatinine		
Magnesium	FEMg	uMg × sCr sMg × uCr	TIF, ATN, CKD, DKD, drug-induced hypoMg	[165,166,167,168,169,170,171,172,173,174]
Osmolality	Estimated	Urine density × 325	Water intake, CKD progression, ADPKD, polyuric syndromes	
	Calculated	2 (uNa + uK) + (UUN/2.8)		[182,183,184,185]
	Measured	Osmometer		
Urine Anion Gap	UAG	(uNa + uK) − uCl	NAGMA, RTA, acidosis by unmeasured anions	[186,187,188,189,190,191,192,193]
	UOG	Measured − Calculated Osm		
pH	pH	Potentiometer	RTA, metabolic syndrome, AKI	[199,200,201,202,203,204,205,206,207,208,209,210,211]
Urine Na^+^ to K^+^ ratio	uNa/uK	uSodium uPotassium	Diet, HBP, AKI, CKD	[212,213,214,215,216,217,218,219,220,221]

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
