# Peer review of "Understanding Renal Tubular Function: Key Mechanisms, Clinical Relevance, and Comprehensive Urine Assessment"

_pathophysiology, 2025, doi:10.3390/pathophysiology32030033_

Round 1

Reviewer 1 Report

Comments and Suggestions for Authors

The manuscript by Drs Alamilla-Sanchez, et al, reviews the use of laboratory studies to examine the function of the renal tubules. The authors correctly point out that most references to "renal function" mean the glomerular filtration rate.

The authors do a very nice job of reviewing the literature in terms of summaries of studies that have evaluated renal tubular function. The reference list is quite extensive and includes most of the pertinent literature.

In addition, the authors have very nicely outlined the underlying physiology of the renal tubules that serves as the basis for the studies.

There are a few areas that I think can be improved.

First, there needs to be more physiology background in the section of the proximal tubule. The authors go through a lot of studies discussing glucose and phosphate, etc, but should start this section with a paragraph or 2 describing the underlying physiology. I would suggest looking at the excellent review article by Dr Rector (AJP-Renal, 1983, pp F461-578). In particular, Figure 7 form that article can help the reader understand some of the discussion regarding glucosuria, phosphaturia and amino aciduria.

Second, the role of the fractional excretion of urea should be discussed. This topic has become very helpful in evaluating patients with congestive heart failure. It has been shown to be predictive of mortality in these patients. It can also be used in evaluating AKI when the patient has been given diuretics such as lasix.

Third, in the section on magnesium, the authors mention calcineurin inhibitors as a cause of magnesium wasting. This is mentioned only briefly in parentheses. This needs to be brought out more since it is a well known cause of hypomagnesiumia.

Comments on the Quality of English Language

The overall quality of the English is good. However, there are a number of typos. It should not require too much work to correct these.

Author Response

Dear Reviewer, we are extremely pleased that the document has received your approval. We value all the observations you have provided and consider them very commendable.

Comment 1: First, there needs to be more physiology background in the section of the proximal tubule. The authors go through a lot of studies discussing glucose and phosphate, etc, but should start this section with a paragraph or 2 describing the underlying physiology. I would suggest looking at the excellent review article by Dr Rector (AJP-Renal, 1983, pp F461-578). In particular, Figure 7 form that article can help the reader understand some of the discussion regarding glucosuria, phosphaturia and amino aciduria.

Response 1: The aforementioned article by Dr. Rector has been a fundamental source of information on tubular physiology for all nephrologists during our training. This contribution is very valuable, and accordingly, we have decided to include the reference in the text as a brief introduction to the physiology of sodium and chloride reabsorption in the proximal tubule. Additionally, the physiology of reabsorption for other solutes (eg; phosphate, uric acid, citrate) in this nephron segment is described in more detail in each subsection; thus, this annotation has been placed before beginning the first subsection (glucosuria).

Comment 2: Second, the role of the fractional excretion of urea should be discussed. This topic has become very helpful in evaluating patients with congestive heart failure. It has been shown to be predictive of mortality in these patients. It can also be used in evaluating AKI when the patient has been given diuretics such as lasix.

Response 2: What is commented on in their observations is true, so it has been decided to add information on fractional urea excretion in patients with acute kidney injury.

Comment 3: Third, in the section on magnesium, the authors mention calcineurin inhibitors as a cause of magnesium wasting. This is mentioned only briefly in parentheses. This needs to be brought out more since it is a well known cause of hypomagnesiumia.

Response 3: As noted, tacrolimus, a drug frequently used in nephrology, causes hypomagnesemia, and its etiology is related to TMRP6 blocking. Further details have been incorporated into the text, specifically related to Dr. Gratreak's research.

Reviewer 2 Report

Comments and Suggestions for Authors

This review article by Alamilla-Sanchez et al. provides a comprehensive overview of the many partial functions of the tubular apparatus, which, in addition to glomerular filtration, is significantly involved in the function of the kidneys to maintain body homeostasis. I agree with the authors that in everyday clinical practice the diagnosis of the various tubular functional impairments or defects is underrepresented in relation to the glomerular filtration rate and their pathogenetic significance is underestimated. The overview article, which initially focuses primarily on pathophysiological aspects, builds bridges to epidemiological observations and clinical studies, which also make it very readable for physicians and nephrologists working at the bedside. The clear structure of the manuscript not only makes it easy for the reader to understand the complex interrelationships, but also enables targeted reading with reference to specific tubular functions, their disorders and diagnostic tools, as shown in Table 1. In my opinion, the paper should be accepted in the present version.

Just a small comment from my side: For a better understanding of Figure 2, it would be very useful if all the abbreviations were explained again in full in the legend, even if the abbreviations are used in the text.

Author Response

Dear reviewer, your contributions and comments are extremely valuable to us; it certainly help improve the context and orientation of the review in accordance with the observations you have so kindly made.

Comment 1:  For a better understanding of Figure 2, it would be very useful if all the abbreviations were explained again in full in the legend, even if the abbreviations are used in the text.

Response 1: 

We sincerely appreciate that you liked the document, and would also like to inform you that the abbreviations were added correctly in the footer of Figure 2. The tubular function should be addressed consistently and specifically to improve the proportion of often underestimated renal diagnoses that could lead to potentially progressive complications that deteriorate renal function over the long term. The document promotes interest in functional assessment among clinical nephrologists, enabling more accurate management and diagnosis.

Additionally, to enhance the clarity of the text, we have requested support to improve the grammatical style in PaperPal.

Reviewer 3 Report

Comments and Suggestions for Authors

Thank you for this valuable review article, although some points need to be fixed

  • The criteria for choosing the article to retrieve the data must be mentioned
  • There were many abbreviations that must be mention by full name, although you add list at the end but not complete one.
  • In line 51 why under high level of vasopressin urine concentration was depress?
  • In line 78 you mention figure 2 first where is figure 1?
  •  In line 236, mention see above. Which part exactly do you mean?
  • in line 640, wrong information is mentioned 
  • r2 in line 807 and many other line refer to what
Comments on the Quality of English Language

The English could be improved

Author Response

Dear reviewer, your contributions provide valuable guidance, and we believe the review has been of interest to you. This is why we conducted a thorough evaluation of your recommendations and made the necessary adjustments requested.

Comment 1: The criteria for choosing the article to retrieve the data must be mentioned

Response 1: We appreciate the reviewer’s insightful comment regarding the criteria for selecting articles for data retrieval. We employed a structured search strategy using a combination of Medical Subject Headings (MeSH) and text keywords related to tubular function and solute handling in the kidney, focusing on articles published since the 2000s. The search terms included: “kidney tubular function” OR “renal tubule physiology”, AND “urinary biomarkers” OR “solute handling” OR “tubulopathy”, AND “acute kidney injury” OR “chronic kidney disease” OR “AKI” OR “CKD”, AND individual solutes and physiological processes such as “glucosuria”, “phosphaturia”, “aminoaciduria”, “uricosuria”, “citraturia”, “magnesuria”, “calciuria”, “urine osmolality”. Classical or foundational studies were included when no recent high-quality evidence was available for specific transport mechanisms.

Articles were further screened by title and abstract, and full texts were assessed for eligibility based on relevance to human kidney physiology and clinical nephrology practice. Classical or foundational studies were included when no recent high-quality evidence was available for specific transport mechanisms.

Comment 2: There were many abbreviations that must be mention by full name, although you add a list at the end but not a complete one.

Response 2: We reviewed the words with abbreviations, and the words in the last section of the article have been adjusted.

Comment 3: In line 51 why under high level of vasopressin urine concentration was depress?

Response 3: In the study "Functional Profile of the Isolated Uremic Nephron" (reference 7), researchers investigated the water permeability and vasopressin responsiveness of isolated cortical collecting tubules (CCTs) from uremic rabbit kidneys. They found that uremic CCTs exhibited impaired water permeability and adenylate cyclase responsiveness to vasopressin compared to normal CCTs. This impairment suggests a defect in the cellular response to vasopressin, potentially at a step following the formation of cyclic AMP. This effect occurs in chronic kidney disease, leading to a decreased tubular response despite high levels of AVP.

Comment 4: In line 78 you mention figure 2 first where is figure 1?

Response 4: We appreciate the observation and would like to mention that this paragraph was not placed correctly; it has been moved just above Figure 2.

Comment 5:  In line 236, mention see above. Which part exactly do you mean?

Response 5: The words "see above" were removed from the text due to the confusion they generate and because they do not add clarity to the text.

Comment 6: in line 640, wrong information is mentioned  

Response 6: As the reviewer has rightly mentioned, there is an error because the order of the factors was inverted. It has been corrected by placing the modifiable factors—weight and 25-O-Hydroxyvitamin D3 intake—next to the non-modifiable factors: ethnicity and menopause.

Comment 7: r2 in line 807 and many other line refer to what

Response 7: “The baseline Uosm was 504 ± 177 mOsm/kg and exhibited a negative correlation with age (r2 = 0.041, p < 0.001) and total kidney volume (TKV) (r2 = 0.044, p < 0.001) and positively with eGFR (r2 = .117, p < 0.001)”  r² indicates the percentage of variability in Uosm that can be explained by each variable (age, TKV or eGFR), showing the strength of the linear relationship: 4.1% by age, 4.4% by TKV and 11.7% by eGFR, all of which are statistically significant (p < 0.001).

Additionally, to enhance the clarity of the text, we have requested support to improve the grammatical style.

Round 2

Reviewer 1 Report

Comments and Suggestions for Authors

The authors have satisfactorily addressed my concerns.